# Getting the message across; a realist study of the role of communication and information exchange processes in delivering stroke Early Supported Discharge services in England

**Niki Chouliara**[1,2]*, **Trudi Cameron**[1], **Adrian Byrne**[1,3], **Rebecca Fisher**[1,4]

**1** School of Medicine, University of Nottingham, Nottingham, United Kingdom, **2** NIHR Applied Research Collaboration (ARC) East Midlands, Nottingham, United Kingdom, **3** School of Computer Science, University College Dublin, Dublin, Ireland, **4** Clinical Policy Unit, NHS England, Nottingham, United Kingdom

* niki.chouliara@nottingham.ac.uk

**Data Availability Statement:** Due to ethical restrictions data are only available to qualified researchers upon request. This study analyses

## Abstract

### Background

Stroke early supported discharge (ESD) involves the co-ordinated transfer of care from hospital to home. The quality of communication processes between professionals delivering ESD and external stakeholders may have a role to play in streamlining this process. We explored how communication and information exchange were achieved and influenced the hospital-to-home transition and the delivery quality of ESD, from healthcare professionals' perspectives.

### Methods

Six ESD case study sites in England were purposively selected. Under a realist approach, we conducted interviews and focus groups with 117 staff members, including a cross-section of the multidisciplinary team, service managers and commissioners.

### Results

Great variation was observed between services in the type of communication processes they employed and how organised these efforts were. Effective communication between ESD team members and external stakeholders was identified as a key mechanism driving the development of collaborative and trusting relationships and promoting coordinated care transitions. Cross-boundary working arrangements with inpatient services helped clarify the role and remit of ESD, contributing to timely hospital discharge and response from ESD teams. Staff perceived honest and individualised information provision as key to effectively prepare stroke survivors and families for care transitions and promote rehabilitation engagement. In designing and implementing ESD, early stakeholder involvement ensured the services' fit in the local pathway and laid the foundations for communication and partnership working going forward.

qualitative data and participants did not consent to have their full transcripts made publicly available. The transcripts contain rich information which could identify patients, staff members and research sites. In line with our agreements with NHS HRA and Hospital R&D Departments the full transcripts of interviews cannot be made publicly available. Instead, supporting de-identified excerpts from the raw data (quotes from interviews with participants) are available within the text.

**Funding:** This study was funded by the National Institute of Health Research (NIHR) Health Services & Delivery Research (HS&DR) project grant (NIHR HS&DR Project: 16/01/17). RF was the budget holder. The research was conducted by the authors and the funder was not involved in the design of the study and collection, interpretation of data nor in writing this manuscript. The views expressed are those of the authors and not necessarily those of the NIHR or the Department of Health and Social Care.

**Competing interests:** The authors have declared that no competing interests exist.

## Conclusions

Findings highlighted the interdependency between services delivering ESD and local stroke care pathways. Maintaining good communication and engagement with key stakeholders may help achieve a streamlined hospital discharge process and timely delivery of ESD. ESD services should actively manage communication processes with external partners. A shared cross-service communication strategy to guide the provision of information along to continuum of stroke care is required. Findings may inform efforts towards the delivery of better coordinated stroke care pathways.

## Introduction

Stroke is a major cause of death and one of the main causes of adult disability [1, 2]. To address stroke survivors' complex needs and respond to pressures for integrated care and cost reductions, models of home-based rehabilitation have gained increasing interest [3]. Early Supported Discharge (ESD) involves the co-ordinated transfer of care from hospital to the patient's place of residence and provision of timely and specialist rehabilitation by a multidisciplinary team (MDT) to enhance recovery [4].

Stroke clinical guidelines in England and worldwide, recommend the provision of ESD as part of an evidence-based stroke care pathway [5–9]. In England, about a third of stroke survivors leave hospital with an Early Supported Discharge (ESD) or community stroke rehabilitation team [10, 11]. Research evidence and a consensus document have provided a good understanding of how ESD should be delivered; ESD should be stroke specialist and responsive, with treatment at home beginning within 24 hours of hospital discharge, offering intensive rehabilitation, based on patients' needs [4, 5, 12, 13]. Clinical guidelines have highlighted the role that ESD teams hold in streamlining the transfer of care from hospital to home and reassuring people with stroke and their family carers [5]. The benefits of ESD have been evidenced in the literature and include reducing the risk of stroke survivors becoming dependent and institutionalised [4, 14]. Questions remain, however, about how evidence-based ESD is implemented at scale, in real-world clinical settings.

The transition of care from hospital requires input from a range of health, social care and volunteer organisations, however, achieving good communication between partners through the continuum of service provision can be challenging. The lack of communication and dialogue between stakeholders has been previously identified as a key barrier to inter-organisational collaboration and integrated healthcare provision [15]. ESD case studies in England and other countries have previously identified deficiencies in information provision and communication between healthcare providers [16–20]. The impact of these communication breakdowns on patient experience has also been described [21, 22]. According to a systematic review investigating stroke survivors' experiences of post-acute care, ineffective communication with healthcare teams resulted in feelings of frustration and abandonment [23]. Acknowledging the complexity of hospital discharge, research and stroke clinical guidelines have highlighted the importance of good communication and timely information transfer between the involved stakeholders in order to achieve safe and seamless transitions and continuity of care [5, 7, 8, 20]. Yet, questions remain regarding how effective communication can be achieved in clinical practice by stroke rehabilitation services.

Understanding how stroke rehabilitation models can be best implemented in the community to enhance stroke survivors' experience of the stroke care pathway, is a priority for the

national health service in England [3]. Policy documents guiding stroke care provision have acknowledged the importance of ESD services being delivered within integrated systems of care [24, 25]. Integrated stroke delivery networks (ISDNs) have been set up to facilitate collaborations between all relevant stakeholders, including patients and relatives, to support joined up, personalised stroke care and promote effective patient flow across the pathway [24]. This reflects an acknowledgment of the importance of communications between service users and the healthcare professionals involved in their care to improve stroke survivors' experience of service provision. Understanding how communication between stakeholders and ESD services can be optimised, will be important to inform these efforts.

The variability of ESD models and delivery networks in England and internationally highlights the need to investigate communication processes in relation to the healthcare contexts in which they are embedded. Rather than approaching communication as part of a list of distinct determinants studied in isolation from local context [26], we wanted to examine how and in what conditions it drives the delivery of seamless transitions and needs-based care. The guidance for conducting and evaluating complex interventions issued by the Medical Research Council (MRC) in England [27], emphasises the need to theorise how and under what contextual conditions complex healthcare programmes work in clinical practice. To support real world decision making, it is suggested that these theories should be informed by the perspectives of healthcare professionals and service users.

Responding to these recommendations, this study aimed to explore how ESD team members achieved good communication and information exchange with key stakeholders, including health and social care professionals and service users, and how these processes influenced the hospital- to -home transition and the delivery quality of ESD in line with evidence-based targets.

## Methods

### Methodological framework

The study was conducted as part of a larger, multi-method realist evaluation project (What Is the Impact of Stroke ESD; the WISE study) designed to assess whether and how evidence-based ESD models were implemented in England [28]. Underpinned by the critical realist philosophy of science, realist evaluation (RE) provides a research methodology for conducting evaluations of complex healthcare interventions, such as community based public health programmes, with wider learning potential [29].

RE involves the development and testing through data collection of hypotheses and theories that describe how a healthcare programme, or certain aspects of the programne, work and in what conditions. Programme theories consist of a series of hypotheses expressed as Context-Mechanism-Outcome (CMO) configurations which outline how the interaction between the intervention's mechanisms (M) and the context (C) in which it is implemented, generate the outcomes of interest (O) [30]. We conceptualised mechanisms as an interaction between the resources offered by the intervention and stakeholders' reasoning and responses to these resources [31, 32]. This line of enquiry permits researchers a better insight into the generative mechanisms and conditions responsible for the success or failure of the intervention; theory informed improvement strategies can then be developed to target barriers and foster the conditions which will help achieve intended outcomes.

The development of our initial programme theory (IPT) was Informed by clinical guidelines and a Rapid Evidence Synthesis (RES) [33]. The aim of the evidence synthesis was to identify barriers and facilitators to the implementation of home-base rehabilitation and highlight features of context we would need to consider as part of our hypothesis generation. Our

IPT suggested that "the quality of communication processes between key stakeholders and services in the local stroke care pathway will influence the delivery of evidence-based ESD". To refine this programme theory, we sought to understand "how" (Mechanisms) and "in what conditions" (Contexts) communication processes drive the delivery quality of ESD services (Outcomes). Outcomes reflected evidence-based quality standards; according to clinical guidelines, key targets for ESD services include ensuring a streamlined hospital discharge and providing responsive, intensive and needs-based rehabilitation at home [5]. Responsiveness, in specific, refers to the number of days from hospital discharge to first face-to-face contact; ESD teams are required to provide treatment at home within 24 hours of hospital discharge and at an intensity similar to what patients would receive in hospital.

The following stage involved testing and refining the programme theory, and associated CMOs, through data collection and analysis. In conducting and reporting the study we followed established quality criteria for realist evaluations [30].

## Sampling and data collection

We identified the study sites using a purposive sampling approach [34]. We selected six ESD services in England aiming to include different geographical areas and capture variation in relation to models of ESD provision and levels of rurality [35], characteristics pertinent to the questions addressed by the wider WISE project. Realist evaluation is method neutral, suggesting that both qualitative and quantitative approaches can be used. In this study, qualitative information was obtained through interviews and focus groups with staff participants.

Key informants from each site were recruited through purposive and snowball strategies. Service managers, team leads and commissioners were invited to take part into one-to-one interviews. We also sought to conduct two focus groups per site with a cross-section of the multidisciplinary team (MDT) staff members, including therapy and nursing staff, rehabilitation assistants, physicians, and administrators. Line manager permission was granted before MDT members were approached for recruitment. The first contact with each service was via phone/email but all interviews and focus groups were conducted in person at each ESD site. Recruitment at each site was completed once our sample captured representation from all different disciplines and roles within each MDT.

Our data collection was in keeping with a realist approach to interviewing [36, 37]. We invited participants' reflections on our initial programme theory, to directly test and refine our hypotheses. Interview schedules were revised in response to important insights that emerged through the data collection process and merited follow-up in subsequent interviews.

One-to-one interviews were conducted by NC (1st author, PhD) and focus groups were conducted by both NC and TC (2nd author, PhD). NC and TC are researchers with substantial experience in collecting and analysing qualitative data from interviews and focus groups. Interviews took place between September 2018 and August 2019, and lasted between 45 minutes and 2 hours (for group interviews). Audio recordings were transcribed verbatim and imported into QSR International NVivo 12 Pro for Windows.

## Ethics approval and consent to participate

Ethical approval was granted by the Nottingham 1 REC Committee (IRAS ID 243066; 18/EM/0160) and the Health Research Authority on 23.07.2018. Written informed consent was obtained by all participants for taking part in this study.

### Data analysis

The data analysis phase was directed towards refining CMOs outlining aspects of the programme theory [38]. An iterative retroductive analytic process [39] was followed which included both top-down (deductive) and bottom-up (inductive) phases.

1. The first phase (deductive) involved reading and coding each transcript using codes informed by our preliminary theory and associated CMOs [40]. Only data pertinent to the research aims were included for analysis.

2. The second phase (inductive) involved reviewing coded excerpts to identify strings of CMO configurations [41]. These where then coded either as triads (C-M-O), when all the three components could be discerned, or dyads (C-M, M-O combinations). These codes were revised and refined as analysis progressed to make the relationship between CMOs clearer and ensure they accurately captured respondents' narratives. Text which could not be coded as a CMO combination, but offered important information was revisited at a later stage of the analysis as connections between CMO features became clearer. Once data coding for each site was completed, cross-site comparisons of key findings were conducted to identify confirming and disconfirming cases and explore how pertinent mechanisms interact with site-specific contextual conditions to generate variation in outcomes. Unexpected findings and data that disconfirmed our initial CMOs, were also coded and contributed to revising and refining our hypotheses.

To minimise ambiguity and promote the trustworthiness of our findings, data were independently analysed by two researchers (NC, TC); any differences in opinion were discussed between the coders and the wider research group and were either resolved or recorded if they reflected informative contradictions in participants' narratives.

The results section that follows, presents key CMOs we identified in relation to our overarching programme theory. They are thematically grouped, based on whether they refer to communication processes between ESD services and: 1) other services on the stroke care pathway, 2) referring hospital-based teams, 2) people with stroke and family members. To better illustrate the link between the CMO tables and the narrative, we have indicated in brackets where in the text features of Context, Mechanisms (Resources or Responses) and Outcomes are described. Quotes are coded based on the Site ID (A to E), the participant ID and their role in the team. Where quotes were taken from focus groups, this is also noted, along with the number of the group (i.e. FG1 or FG2).

## Results

In total we spoke to 117 staff members directly involved with ESD delivery (Table 1). No respondents withdrew participation at any point.

### Communication across the pathway

Hospital discharge requires the coordinated response of health and social care professionals from both hospital and community-based services. Respondents often spoke of fragmented care pathways (Table 2) which made efforts to provide seamless, needs-based care more challenging but also necessary to improve patient experience *(Context)*. According to our participants, ESD services have a key role to play spanning occupational and organisational boundaries (*Mechanism-Resource*) to facilitate communication between involved parts (*Mechanism-Response)* and contribute to the provision of streamlined, needs-based care(*Outcome*).

**Table 1. Break down of staff participants per site and interview method.**

| Site | Interviews(n) | Focus group participants (n) | Total staff participants (n) |
|---|---|---|---|
| A | 5 | 12 | 17 |
| B | 5 | 15 | 20 |
| C | 8 | 9 | 17 |
| D | 2 | 13 | 15 |
| E | 9 | 16 | 25 |
| F | 6 | 17 | 23 |
| Total | 35 | 82 | 117 |

A and B sites offered both ESD and longer-term community rehabilitation services. Sites A and B covered an urban geographical area, sites C and D a semi-rural area and sites E and F a rural area.

*"I have patients phone up and say, "I need to contact the district nurse" so we tend to be kind of like a 118 service (laughs.) It's communication and it's being able to say "I don't know but I'll find that out". And I've had people come back to me at the end saying, "Oh, I'm so glad you're on the end of the phone.""*

*(Site E S17, operational coordinator)*

Participating sites liaised with a number of other services including GPs, hospital services and voluntary organisations to ensure patients' complex needs were recognised and addressed (*Mechanism-Resource*). In some cases, particularly where patients relied on ESD for support, staff would be called to communicate patients' wishes to other services or help them "make sense" of the information they were given (*Mechanism-Response)*, supporting patients to navigate the complexity of care pathways *(Outcome)*.

*"Discharge letters themselves can be quite confusing, so I had to sit and study a discharge letter the other week that said a certain drug had to be stopped. I phoned up the consultants to get clarification so that the patient knows what they're doing."*

*(Site F, focus group 2)*

Good communication and co-ordination with social care services was especially difficult to achieve, leading to unnecessary delays and disjointed transfers. In one site, patients were seen

**Table 2. CMOs on communications between ESD services and local stroke care pathways.**

| Context | Mechanisms | | Outcomes |
|---|---|---|---|
| | Resources | Responses | |
| Complex and multifaceted impact of stroke requiring a coordinated response from multiple services<br><br>Fragmented stroke care pathways | ESD playing a boundary spanning role liaising with a number of services and advocating for patients' needs | Facilitating communication and flow of information across services and between healthcare professionals and service users | Contributing to the provision of streamlined, needs-based care<br><br>Supporting service users in navigating complex healthcare pathways |
| Lack of shared /joined IT systems<br><br>Policy and commissioning changes influencing ESD implementation and delivery | Early engagement with multiple stakeholders, including service users | Securing stakeholders' buy in and involvement in designing the service | Ensuring relevance of ESD model to local context and promoting partnership working |

by multiple private care agencies and staff talked about their efforts to maintain meaningful communication and establish a common approach and ethos towards patients' care.

> *"They are all different agencies. They'll have different standards and different ways of doing things. So we could go in and we may provide rehabilitation and try and promote independence, but the carers come in and they just do it for them. We don't have that communication and these links with them to facilitate rehabilitation".*

> *(Site A S1, Speech and language therapist)*

Participants described a shifting landscape in community stroke rehabilitation provision, impacted by commissioning and policy developments in stroke care in England. Frequent restructuring and decommissioning of services were not always communicated effectively or were identified by respondents as disrupting existing communication processes in the pathway. Lack of shared or joined up information systems across services and organisations further aggravated breakdowns in communication, complicating the collection and exchange of information across services *(Context)*.

> *"I think there has been a lot of talk about electronic notes; they (services that use them) can link directly to GPs, they can communicate across teams using this system and we are nowhere near that. We are still on paper, having to make phone calls to talk to a GP or make a referral to a District Nurse. So from a technology perspective, that can be an issue, I can sit on the phone for 45 minutes waiting to make a referral, again that's not good use of my time."*

> *(Site C S5, occupational therapist)*

Some sites had established processes to streamline communication with other services on the pathway, including regular service management meetings, or "pathway redesign and improvement meetings". Team leads and service managers drew attention to the need for early engagement of multiple stakeholders from across the pathway, including GPs, community and hospital representatives and commissioners *(Mechanism-Resource)* to secure their buy-in and invite their involvement in designing and developing the service *(Mechanism-Response)*. These early exchanges provided a platform for deliberation between stakeholders, helped ensure the relevance of the service model with the local context and laid the foundation for ongoing communication and partnership working *(Outcomes)*. One service highlighted the importance of including service users in these discussions, people with stroke and family members, whose feedback could be valuable in shaping a "realistic" and inclusive service.

> *"I think if you're setting up a service, it's important to go out and speak to people that live their lives. They don't live your life. They live a different life than you have. You've got to have a realistic service. You've got to think about everyone, not just what you'd like to do. Don't just talk to the clinicians. We did a huge amount of work, sought feedback from patients who were invited back to the executive boards to tell their story to the commissioners about what they thought worked well and didn't work well. That's been ongoing from day one throughout. So, what we're now working on is that one stroke pathway with collaborative working across all of us. It's a very collaborative approach and it feels much more positive and less them and us"*

> *(Site F S4, clinical lead for community trust)*

## Communication with referring hospital-based services

The importance of good communication and a collaborative relationship with teams from the referring stroke units was recognised by all participating teams (Table 3). Communication with inpatient services was occasionally hampered by historic, cultural and organisational differences *(Context)*. Participants agreed that communication with referrers needed to be proactively managed, however, great variation was observed between participating teams in the communication processes they had in place. These, most commonly, involved boundary-spanning working arrangements and roles. Where ESD and inpatient services operated under the same organisation, staff rotations were encouraged as well as opportunities for acute staff to "shadow" community-based staff. These arrangements served an educational purpose, promoting clarity about the ESD role and remit and they provided opportunities for formal and informal communication *(Mechanism-Resource)*. Mutual trust and rapport were developed in response as well as referrers' confidence in the quality of specialist care ESD could offer *(Mechanism-Response)*. This contributed to streamlining the discharge process by facilitating the timely referral of patients who met the ESD eligibility criteria, and promoted ESD responsiveness *(Outcomes)*.

> *"The staff have had to be very good at communication and developing strategies to do that. Developing communication but also developing rapport and, I can't think of the right word, I suppose trust as well, communication and trust. Because if the therapists don't trust the community team, then they are reticent to let that patient out so you have to develop that. That has developed over time with communication and just working together and we have some staff who work across all of the pathways in terms of physiotherapy."*

(Site E S18, head of physiotherapy)

Four of participating teams, in-reached into the inpatient rehabilitation beds to identify appropriate referrals and assist with discharge planning. Respondents had mixed views regarding in-reaching; some ESD teams viewed this process as an opportunity for communication and information exchange with patients, families and staff involved in the discharge process. In other sites, this model was trialled but found to be resource intensive and, therefore, abandoned in favour of other processes such as periodic visits to the stroke unit.

> *"I would say the fact that we in-reach is the reason why we get the right patients referred. And I'll do joint sessions on the ward with the nurses and the OT's and speech and language therapists. I think it builds up a rapport. Making sure there's continuity of care, making sure that*

**Table 3. CMOs on communication between ESD services and referrers.**

| Context | Mechanisms | | Outcomes |
|---|---|---|---|
| | Resources | Responses | |
| Historic and cultural differences between hospital and community-based service provision | Boundary spanning roles/ working arrangements: | a. Building mutual trust between ESD services and referrers and promoting clarity about ESD | a. Contributing to the timely discharge of appropriate referrals |
| Different service models (e.g. in reaching/out reaching) requiring different types of communication processes with referrers | a. Providing opportunities for formal and informal communication and knowledge transfer | b. Preparing ESD services to receive discharged patients and respond to their needs | b. Promoting ESD responsiveness |
| High staff turnover in referring teams | b. Facilitating the timely communication of the discharge plan | | |

*everything is okay for the patient when they come home. And that they understand about the service, and they are up for it really. And that all their hopes and wishes and our hopes and wishes coincide."*

*(Site E S5, Physiotherapist)*

Communication of the discharge plan in a timely manner (*Mechanism-Resource*) was considered key to permit ESD teams to prepare for admitting the patient and coordinate a prompt response (*Mechanism-Response*). When this was not achieved, miscommunications between the acute and ESD teams posed a threat to patients' timely discharge and the ability of ESD to meet responsiveness targets *(Outcomes)*.

*"And then it might be patients aren't discharged but nobody thinks to tell us. They've had a delay because of social care, the carers can't start or somebody was meant to bring a bed downstairs and they haven't, so they're still waiting in hospital. But that communication doesn't always come through. So I've got one of our team banging on their (patient's) door trying to do a safety check and they're not even there, so then that's a wasted slot. So that's an inefficiency which we try and minimise by communication, there is no other way. But it's an ongoing battle, it's not easy."*

*(Site D S7, team lead)*

High hospital staff turnover necessitated sustained efforts to maintain the links and educate referrers about ESD. In most cases, a reactive approach was taken, with processes being put in place to remediate miscommunication with referring services. One team, however, discussed the benefits of proactively forging these links at an early stage as a way to encourage collaboration. To achieve this, their service manager visited the referring stroke unit daily for about a year prior to the ESD service's full operation:

*"When we started, we pushed hard to make sure that I went and spend time with the acute unit . . .so we all know each other and have that trust. When you ring up, one end or the other, you know who you are speaking to. So yes, we do have quite good relationships."*

*(Site C S1 &S2, clinical team lead & clinical specialist physiotherapist)*

## Communication with patients/carers

Returning home from hospital was recognised as a critical point in the patients' journey through the care pathway, initiating a process of readjustment and adaptation as patients and families get a better appreciation of how stoke has impacted their lives (Table 4). ESD teams were called to address concerns about leaving the protected hospital environment and ease the transition back home *(Context)*:

*"I think that link between hospitals and home is really important. Because normally patients are in a complete state, they're in shock and they're frightened, and they're worried. And if they can speak to somebody who says, "We're gonna come and see you tomorrow" its important. And if they have any more worries or concerns, we say "What is it we need to do to support them? We try and ensure that the patients understand the service and want it".*

*(Site E S4, nurse)*

**Table 4. CMOs on communication between ESD services, patients and families.**

| Context | Mechanisms | | Outcomes |
|---|---|---|---|
| | Resources | Responses | |
| Patients' and carers' worries and concerns about returning home | Timely, honest and tailored communication between healthcare professionals, patients and families through appropriate formats/channels | Patients and families developing realistic expectations about recovery, and clarity regarding the role of ESD | Setting realistic and relevant rehabilitation goals is facilitated |
| Coming to terms with the impact of stroke | | Families feeling actively involved in rehabilitation | Patients and families are better prepared for care transitions |
| Unmet information needs | | | |
| Patients and referrers' misconceptions regarding the role of ESD | | | Patients' and families' rehabilitation engagement is promoted |

Gaps in information provision prior to discharge often meant that patients arrived home not knowing what to expect and uncertain about the type and level of support they would receive. This was most commonly reported by ESD services that did not "in-reach" into the referring hospital but first met their patients at home. At times ESD teams were called to manage misinformation or misconceptions about what ESD could offer which often reflected referrers' own lack of understanding of the ESD role and remit *(Context)*.

*"She (patient) went home feeling very nervous because the staff on the ward had expressed their concerns. And that was around not appreciating what could be offered to her at home and the level of therapy that she would be able to get at home. If patients are given the wrong expectations for community services, you're heading for a problem."*

*(Site B S18, head of physiotherapy)*

Ongoing pressures to reduce the length of hospital stay were identified by some participants as a factor hindering hospital staff efforts to respond to patients' information needs *(Context)*. ESD staff saw it as part of their role to educate patients about life after stroke and how rehabilitation could support recovery. Staff across sites agreed on the importance of timely and bidirectional communication with patients and family carers *(Mechanism-Resources)* to effectively prepare them for care transitions and support them in developing clarity about the role of ESD and realistic expectations of recovery *(Mechanisms-Responses)*. Being honest and pragmatic *(Mechanism-Resources)* was highlighted by staff as a particularly important feature of their communications with patients and relatives. It was envisaged that this, in turn, enabled setting achievable, patient-centred rehabilitation goals and promoted engagement with rehabilitation regime *(Outcomes)*.

*"I know people don't like being honest and straightforward, but we have to be." And that's what I'm trying to say to my team. With my patients, I have to be honest. I don't do magic, I am honest, but I guarantee you that we will work hard. Not just me, but you. And the patients say why must we? Because I say we have limited resources and you have a really good potential, but we have to have be very strict."*

*(Site A S9, service manager)*

The need to establish communication and rapport with families and family caregivers was stressed by respondents *(Mechanism-Resources)*. Staff agreed on the important role that family members can play towards the success of rehabilitation efforts. Including them in communications was seen as a way to ensure they felt actively involved in patients' rehabilitation journey *(Mechanisms-Responses)*. Considering the time limited input of the ESD service, maintaining these links throughout patients' stay with the service, enabled staff to prepare families for when ESD ended *(Outcomes)*.

*"We involve the carers from day one. We'd ring the family and introduce ourselves and explain our involvement and what the service is about. We'll give them a carers' information pack and we'll ask them if they've got any questions.*

*(Site B, focus group 1)*

One size did not fit all and some respondents commented on the need to employ appropriate communication channels and formats, tailored to the needs of each patient *(Mechanism-Resources)*. They suggested developing customised materials that consider stroke related cognitive and language difficulties and address information needs without overwhelming patients.

*"We use the Stroke Association leaflets. We try not to overwhelm people. In the early days of the service we thought we were being super brilliant by putting together loads of information and we got a lot of feedback from people saying, "Please don't do it, we can't cope with it all." And literally it was just ending up in drawers. So we try to be more selective about what we do."*

*(Site E S1, team lead)*

## Discussion

This study sought to understand how ESD team members achieved good communication and information exchange with key stakeholders and how these processes influenced the hospital-to-home transition and the delivery quality of ESD. Interviews highlighted the importance of effective communication for nurturing collaborative and trusting relationships between stakeholders, and promoting coordinated and seamless care transitions, supporting our initial programme theory. Participants' narratives reiterated the interdependency of ESD services and local pathways; their work cannot be performed in isolation but good communication with external stakeholders was perceived to be key in achieving a smooth hospital discharge and a timely response from ESD. These findings support and augment evidence from smaller scale case studies [17–20]; the methodological approach we followed permitted contextualising the role of communication and allowed a better understanding of how it drives the outcomes of interest.

Although most of the participating services had a well-established position in their local pathways, ongoing efforts to clarify the ESD role and build liaison with referrers were considered necessary. The need to address knowledge gaps among hospital staff regarding the ESD remit and benefits has been stressed in the literature [20, 23]. Referrers' uncertainty about whether and how ESD can effectively support patients at home has been highlighted as one of the main reasons for the service being underutilised or receiving inappropriate referrals. This study found that the formal and informal communication opportunities created by cross-

boundary working arrangements and specialist discharge roles, promoted the development of a trusting relationship and built referrers' confidence around the quality of care provided by ESD. Previous research has commented on the positive functions of these boundary spanning roles which include facilitating the flow of information between care providers [42], de-emphasising differences, and directing the focus onto collective problem solving and collaborative ways of working [43, 44]. Our findings reiterate the value of protecting and further developing opportunities for cross-boundary working to promote communication and collaboration across services.

The position of ESD in the pathway and the timing of their intervention meant that staff members were required to liaise with and signpost their patients to a number of services spanning health, social, and third sector organisational boundaries. ESD staff assumed a knowledge brokering role in order to clarify, "translate" and effectively communicate information from other services back to their patients. Participants' accounts suggested that ESD services may have an important role to play beyond the delivery of active rehabilitation, and towards leading efforts to provide comprehensive, needs-based care to stroke survivors [5]. Current policy in England recommends the delivery of ESD as part of integrated, patient centred, community stroke service models that offer seamless access to a range of services including return to work, carer support services, psychological and neuropsychological services [26].

Effective communication was found to be influenced by early efforts to secure stakeholders' engagement at a pre-implementation stage. Against a background of cultural and organisational divisions, early and inclusive communication with key stakeholders, including service users, clinicians and policy makers helped secure their support and facilitate the implementation of a service relevant to local needs. The importance of involving both the professionals and service users in designing and developing a healthcare programme, has been previously emphasized as a way to encourage a sense of co-production over the innovation and improve its acceptability, reach and sustainability [45]. Co-designing pathways requires finding common avenues for thinking and working together, creating the conditions for good communication and partnership working moving forwards [43].

Hospital discharge has been deemed the most stressful transfer of care for people with stroke and their carers, accompanied by feelings of fear and abandonment [5]. Findings suggested that honest, timely and individualised communication and information provision to both stroke survivors and their families may help address fears and concerns about returning home, and foster adherence and engagement with the rehabilitation regime. As part of the wider (WISE) project, service users' experiences of ESD were sought [46]. Similar to staffs' accounts in this study, people with stoke and their family carers highlighted gaps in information provision prior to hospital discharge, which caused uncertainty about what to expect when they first entered ESD [46]. This is a recurrent theme in our research [22] and the wider literature [23]. According to patient participants however, the issue persisted while under the care of ESD with some patients being unsure of the role of the service, how it differed from other services on the pathway and what would follow their discharge from ESD [45]. Questions are, therefore, raised about whether the communication processes reported by staff in this study were adequate and effective in reaching all their patients.

Our study comes a decade after a consensus document on implementing community stroke rehabilitation services called for a communications strategy to be produced to guide information provision to all patients and carers following transfer from hospital [47]. Our participants agreed on the value of good communication between healthcare professionals and service users, yet services varied greatly in relation to the type of communication processes they had in place and in how coordinated and organised these efforts were. Although a standardisation of communication strategies may not be feasible or appropriate, considering the great

variation in ESD models, our findings suggest that good communication with stakeholders, including patients and their carers, needs to be proactively managed and planned in a timely manner and in collaboration with referring teams.

Albeit a resource intensive process, using a realist approach to analysing qualitative data is suggested to promote the explanatory power of findings [48]. Rather than studying the influence of communication in isolation, RE methodology allowed us to examine it as an integral part of the process through which intervention components interact with actors' responses and contextual conditions to lead to the outcomes of interest [27]. A limitation of the approach is the ambiguity that may characterise the analytical task of distinguishing between features of context and intervention mechanisms; we found that conceptualising mechanisms as an interaction between intervention resources and actors' responses facilitated the process of identifying CMO chains.

Applying the RE quality standards helped us assure rigour in conducting and reporting the study. The realist approach to interviewing allowed us to test our own assumptions with study participants, promoting the trustworthiness of the findings. The validity of our interpretations and the relevance of key findings to their local settings was further assessed through discussions with participating teams over two feedback workshops. Although not aspiring to quantitative generalisability, following a theory driven enquiry is said to strengthen the transferability of the findings to other settings with comparable characteristics. Our data came from a sample of ESD services in England; the observed variation in ESD models of operation within the country suggests that the insights generated by this data should be further tested in different contexts in order to develop a higher-level mid-range theory of communication processes in early supported discharge services.

Our findings map onto key components of the normalisation process theory (NPT) [49] as they highlight the need for stakeholders to understand the role, remit and function of ESD (coherence or sense making construct), and stress the importance of developing networks (collective action) and nurturing engagement (cognitive participation) in order to "normalise" ESD in the local care pathways. As such, we suggest that NPT may be of value as a middle range theory when examining the role of communication and knowledge transfer in the delivery of complex community-based healthcare interventions.

With regards to the study of contextual influences to the implementation of ESD schemes, our findings point to the need to understand inter-organisational and system level factors in addition to team and service level determinants. The concept of "Cosmopolitanism" described in implementation frameworks [50, 51] is aligned with our observations of the services' networking efforts through boundary spanning communication processes. Stakeholder representation from across the stroke care pathway in future research, may allow a better understanding of how system level factors shape ESD implementation and delivery. The rurality of their location informed site selection in this study. It has been suggested that rural services require additional resources to provide responsive and intensive ESD [52]. In their interviews, staff members participating in this study suggested that rurality had a negative impact on internal communications between the MDT members (reported elsewhere [33]). However, the same respondents, did not identify rurality as a factor influencing external communications and information exchange.

In conclusion, our findings highlighted the need for ESD teams to proactively manage communication processes with key stakeholders with the view to supporting the provision of a seamless and patient centred service. Working collaboratively with referring services to develop a shared communication strategy may be required to meet patients' and families' information needs in a timely fashion. Creating opportunities for early engagement and ongoing formal and informal communication may help clarify the role of ESD in the care pathway

and set the ground for a more coordinated and integrated way of working between key partners.

## Supporting information

**S1 File. Interview schedule.**
(DOCX)

**S2 File. Focus group guide.**
(DOCX)

## Acknowledgments

The authors acknowledge and thank the NHS staff and stroke survivors who participated in this study. We also thank Trevor Gard and Frances Cameron, stroke survivors, who were members of our study steering group.

## Author Contributions

**Conceptualization:** Niki Chouliara, Rebecca Fisher.

**Data curation:** Rebecca Fisher.

**Formal analysis:** Niki Chouliara, Trudi Cameron.

**Funding acquisition:** Rebecca Fisher.

**Investigation:** Niki Chouliara, Adrian Byrne.

**Methodology:** Niki Chouliara, Rebecca Fisher.

**Project administration:** Rebecca Fisher.

**Resources:** Rebecca Fisher.

**Supervision:** Rebecca Fisher.

**Validation:** Trudi Cameron, Adrian Byrne, Rebecca Fisher.

**Writing – original draft:** Niki Chouliara, Trudi Cameron, Adrian Byrne, Rebecca Fisher.

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
