## [Decision Letter · Decision Letter 0]

26 Oct 2023

PONE-D-23-14195Getting the message across; a realist study of the role of communication and information exchange processes in delivering Stroke Early Supported Discharge services in England.PLOS ONE

Dear Dr. Chouliara,

Thank you for submitting your manuscript to PLOS ONE. After careful consideration, we feel that it has merit but does not fully meet PLOS ONE’s publication criteria as it currently stands. Therefore, we invite you to submit a revised version of the manuscript that addresses the points raised during the review process.

We look forward to receiving your revised manuscript.

Kind regards,

Maria Berghs, PhD

Academic Editor

PLOS ONE

Journal Requirements:

Additional Editor Comments :

Thank-you for your patience with this review. The reviewers were both largely positive and gave constructive advice Reviewer 2, in particular, recommends some more details to ensure methodological rigor which are very helpful.

Reviewers' comments:

Reviewer's Responses to Questions

**Comments to the Author**

1. Is the manuscript technically sound, and do the data support the conclusions?

Reviewer #1: Yes

Reviewer #2: Yes

2. Has the statistical analysis been performed appropriately and rigorously? 

Reviewer #1: N/A

Reviewer #2: N/A

3. Have the authors made all data underlying the findings in their manuscript fully available?

Reviewer #1: Yes

Reviewer #2: No

4. Is the manuscript presented in an intelligible fashion and written in standard English?

Reviewer #1: Yes

Reviewer #2: Yes

5. Review Comments to the Author

Reviewer #1: This is an interesting and valuable piece of work. It addresses a much needed area of healthcare given the current conditions in the NHS and the sometimes failure of integrated health and social care services.

The background offers thorough and comprehensive context drawing on both literature and policy. The discussion and reflection on the use of Realist evaluation at both the methodological point and in the discussion means that this paper can serve as a useful paper for any new to the process. I thought that this was particularly welcome.

I was expecting to see CMOS set out early on but actually the integration of each set for each finding was very effective. The reader can clearly see how these have been realised or not. The use of quotes peppering this section adds to the reader's understanding (rather than have them set aside in a table as is sometimes the case).

The discussion is extensive. At first I felt perhaps a little too much but actually it is so well informed and makes very specific and pertinent links between the findings and other studies. I liked the way that this was very methodically undertaken. As referred to above, the reflection on RE is a helpful one. However, the later paragraph on normalisation theory, while important, felt like an additive. I wonder whether some mention of this at an earlier point would be useful.

A final important point to make is that I think the findings could clearly have impact on practice, and this makes it an important piece of work. It will also be a valuable paper for anyone undertaking realist evaluation, especially as a novice.

I would highly recommend publication.

Reviewer #2: Very interesting and well written paper.

Reference the RAMESES guidelines for realist evaluation.

Clarify if a literature or realist review was conducted in advance of the realist interviews. The authors state that the work presented is part of a larger realist method and then state that a literature review (para three methodological framework section) was conducted. Clarify this point in the context of the realist methods / realist study presented in this paper.

Add some details regarding the realist informed interview questions for the interviews / focus groups.

Clarify if one or more initial programme theories were developed, my understanding is that it was one IPT relating to communication. Include a clearer statement of the initial programme theory with CMO configuration as an if/then statement in para three of the methodological framework section.

Methodological framework section, last line, reference the RAMESES guidelines.

Sampling and data collection section, four para, states theory / theories, one IPT tested in this work?

Clarify that coding was in accordance with the IPT and therefore 'theory coding' as part of the analysis.

Table 2, add some more details to the statements for example, building mutual trust for whom / by whom etc, ensure that there is sufficient depth to the statements in the context of the CMO configuration and IPT. Review all tables to ensure sufficient granulation of details.

Discussion, the authors introduce an interesting point regarding cosmopolitanism, in the theorising, consider adding more detail here to deepen the discussion in accordance with realist evaluation.

6. PLOS authors have the option to publish the peer review history of their article (what does this mean?). If published, this will include your full peer review and any attached files.

Reviewer #1: **Yes: **Jane Montague

Reviewer #2: No

---

## [Author Response · Author response to Decision Letter 0]

16 Dec 2023

Reviewer 1 

 This is an interesting and valuable piece of work. It addresses a much needed area of healthcare given the current conditions in the NHS and the sometimes failure of integrated health and social care services.

The background offers thorough and comprehensive context drawing on both literature and policy. The discussion and reflection on the use of Realist evaluation at both the methodological point and in the discussion means that this paper can serve as a useful paper for any new to the process. I thought that this was particularly welcome.

I was expecting to see CMOS set out early on but actually the integration of each set for each finding was very effective. The reader can clearly see how these have been realised or not. The use of quotes peppering this section adds to the reader's understanding (rather than have them set aside in a table as is sometimes the case).

The discussion is extensive. At first I felt perhaps a little too much but actually it is so well informed and makes very specific and pertinent links between the findings and other studies. I liked the way that this was very methodically undertaken. As referred to above, the reflection on RE is a helpful one. However, the later paragraph on normalisation theory, while important, felt like an additive. I wonder whether some mention of this at an earlier point would be useful.

A final important point to make is that I think the findings could clearly have impact on practice, and this makes it an important piece of work. It will also be a valuable paper for anyone undertaking realist evaluation, especially as a novice.

I would highly recommend publication.

Author Response:

We would like to thank the reviewer for their comprehensive and very positive feedback. 

Reviewer 2 

Reviewer 2.1.:Very interesting and well written paper.

Response: Thank you for the positive feedback. 

Reviewer 2.2: . Clarify if a literature or realist review was conducted in advance of the realist interviews. The authors state that the work presented is part of a larger realist method and then state that a literature review (para three methodological framework section) was conducted. Clarify this point in the context of the realist methods / realist study presented in this paper.

Response: Thank you, we have now clarified the type of review and how it contributed to the development of our IPT. 

Reviewer 2.3: Add some details regarding the realist informed interview questions for the interviews / focus groups.

Response: We have now included the interview schedules as supplementary files. 

Reviewer 2.4.Methodological framework section, last line, reference the RAMESES guidelines.

Response: Reference number now corrected 

Reviewer 2.5. Sampling and data collection section, four para, states theory / theories, one IPT tested in this work?

Response: Thank you, this study was part of a wider project which tested further programme theories. However, this paper focuses on one of them and hence we rephrased to avoid confusion. 

Reviewer 2.6. Clarify that coding was in accordance with the IPT and therefore 'theory coding' as part of the analysis.

Response: In line with recommendations for realist evaluations we followed a retroductive analytic process which included both top-down (theory informed) and bottom-up (inductive) phases. We begun the analysis by assigning codes directly informed by our IPT but we then proceeded to code data that “disconfirmed”, challenged or added further depth and detail to our preliminary theory and could, therefore, inform revisions/refinements. 

Reviewer 2.7. Table 2, add some more details to the statements for example, building mutual trust for whom / by whom etc, ensure that there is sufficient depth to the statements in the context of the CMO configuration and IPT. Review all tables to ensure sufficient granulation of details. 

Response: Table 2 focuses on communication between ESD services and referrers. This information is now repeated in the table to ensure it is clear to the reader.

---

## [Editor Report · Decision Letter 1]

21 Jan 2024

Getting the message across; a realist study of the role of communication and information exchange processes in delivering Stroke Early Supported Discharge services in England.

PONE-D-23-14195R1

Dear Dr. Chouliara,

We’re pleased to inform you that your manuscript has been judged scientifically suitable for publication and will be formally accepted for publication once it meets all outstanding technical requirements.

Kind regards,

Maria Berghs, PhD

Academic Editor

PLOS ONE
---

## [Editor Report · Acceptance letter]

29 Feb 2024

PONE-D-23-14195R1 

PLOS ONE

Dear Dr. Chouliara, 

I'm pleased to inform you that your manuscript has been deemed suitable for publication in PLOS ONE. Congratulations! Your manuscript is now being handed over to our production team.

Kind regards, 

on behalf of

Dr. Maria Berghs 

Academic Editor

PLOS ONE